# Healthcare Service Gap Analysis: A Comparison of Trend Data from 2018 and 2022 Dubai Clinical Services Capacity Reports

**DOI:** 10.3390/healthcare13172127

**Published:** 2025-08-27

**Authors:** Nahed Monsef, Elham Ashkar, Meenu Soni

**Affiliations:** 1Strategy and Governance Department, Dubai Health Authority, Dubai P.O. Box 4545, United Arab Emirates; namonsef@dha.gov.ae; 2Health Economics & Insurance Policies Department, Dubai Health Authority, Dubai P.O. Box 4545, United Arab Emirates; mschandersoni@dha.gov.ae

**Keywords:** Dubai Clinical Services Capacity Plan, capacity plan, supply-demand analysis

## Abstract

**Background:** Dubai’s healthcare system is designed to meet the growing needs of its population while maintaining high standards of accessibility, quality, equity, and responsiveness. The Dubai Health Authority (DHA) uses planning tools to assess residents’ health requirements and implement effective regulatory strategies. This study compares trend data from the 2018 and 2022 Dubai Clinical Services Capacity Plan (DCSCP) reports to understand how population changes have impacted healthcare demand and to identify service gaps addressed over four years with particular focus on key medical specialties in high demand. **Methodology:** This study retained the methodologies used in the 2018 and 2022 Dubai Clinical Services Capacity Plan (DCSCP) reports, capturing healthcare supply through a census of licensed facilities in Dubai and estimating demand using a need-based approach aligned with diagnosis-related groups (IR-DGRs). The data is categorized into eight key health-planning units (KPUs) to highlight gaps across major service categories and assess whether these gaps have been resolved in a rapidly evolving healthcare system. **Results:** Between 2018 and 2022, there were clear improvements across several key planning units. The shortage of acute overnight beds was resolved, moving from a deficit of 239 beds to a surplus of 1728 beds, an overall gain of 1967 beds. Outpatient consultation rooms also saw major growth, shifting from a gap of 1769 rooms in 2018 to a surplus of 4707 rooms in 2022, a net increase to 6476 rooms. In addition, emergency department capacity increased, and the number of ICU beds also rose from 484 to 691, an overall growth of 43%. These changes represent measurable improvements in acute care and outpatient service capacity. However, despite the addition of 76 beds, long-term care continues to show a shortfall of 138 beds, indicating that this remains a significant gap in Dubai’s healthcare system. **Conclusions:** Dubai has made significant progress in expanding its healthcare infrastructure between 2018 and 2022, addressing many capacity shortfalls, particularly in critical care and outpatient services. However, challenges in non-acute and long-term care remain, requiring ongoing strategic planning to meet future healthcare needs.

## 1. Introduction

Strategic healthcare capacity planning is a cornerstone of resilient health systems, enabling policymakers to align infrastructure, workforce, and service delivery with evolving population needs. While countries with established health systems—such as the United Kingdom, Germany, and Singapore—routinely use population-based projections, diagnosis-related group (DRG) modeling, and demand forecasting to balance supply and utilization, there is limited research on how rapidly expanding, demographically unique cities implement comparable methods. This gap is particularly relevant for global health planners as urban centers worldwide face increasing pressures from migration, medical tourism, and private-sector dominance, factors that challenge traditional public health planning models. This study seeks to explain how Dubai’s healthcare capacity evolved between 2018 and 2022, and whether this methodology can provide a transferable model for health systems facing rapid growth and complex demographics. The DCSCP is a periodic study conducted to understand the current healthcare sector’s demand, supply, and existing service gaps. The study was conducted across government and private sectors in Dubai, and includes all healthcare facilities operating within the geographical jurisdiction of the Emirate of Dubai [1]. Dubai’s healthcare system is rapidly evolving and represents a fast-growing healthcare market, with a reported 3431 healthcare facilities—including hospitals and clinics—in 2018 increasing to 4482 in 2022. There has been significant growth in hospitals and outpatient facilities leading to an increase in capacity for the defined KPU specialties, medical equipment, and healthcare workforce.

Globally, many advanced health systems employ similar strategic capacity planning approaches to balance supply with rising demand. For instance, countries such as the United Kingdom and Australia use population health projections and disease burden modeling to inform regional service distribution and infrastructure investment. Germany and Singapore integrate predictive analytics and DRG-based funding to optimize acute care and shift utilization towards day-case services, reducing overnight bed dependency. In the United States, states like California use certificate-of-need frameworks to regulate healthcare infrastructure expansion, ensuring alignment with population needs and controlling oversupply. These examples demonstrate that while methodologies differ, strategic forecasting, workforce modeling, and targeted investment are widely adopted to build resilient systems. Framing Dubai’s approach alongside these international practices underscores the relevance of the DCSCP as a globally aligned tool for healthcare system sustainability and adaptability.

Regarding population structure, the Dubai population is a young, predominately male population, and largely expatriate. By the end of 2022, Dubai’s population was estimated at approximately 3,549,900 individuals, with males accounting for 68.7% and females for 31.3% of the population [2]. According to the 2022 DCSCP report, Dubai’s population structure is relatively young, with males comprising 69% (2,450,176) and females 31% (1,100,696) of the total, with a significant portion of the population—91%—comprising non-nationals or expatriates [2]. The high prevalence of chronic and non-communicable diseases among the national population underscores the growing need for healthcare services, specialized tertiary care, and chronic disease management. Cardiovascular diseases remained the leading cause of mortality in Dubai in 2022, accounting for 41% of total deaths [3].

The need for investment in healthcare services is driven by increasing demand, influenced by several key factors including population growth, rising utilization of healthcare services, the prevalence of chronic diseases, insurance coverage, and health tourism [4].

Between 2018 and 2022, Dubai’s healthcare sector was dominated by the private sector regarding both inpatient and outpatient volume compared to the public sector. In 2022, 73% of outpatients and 83% of inpatients were treated in private healthcare facilities. The private sector also saw a significant increase in inpatient beds, with a compound annual growth rate (CAGR) of 15.5%, while the public sector experienced a CAGR of 6.8%. In addition, Dubai ranks No. 1 in the Middle East and North Africa and No. 6 globally for medical tourism [4], its rise being driven by high standards, strict regulations, and world-class healthcare facilities with cutting-edge technology [5]. The main goal of the DCSCP reports is to evaluate patient demographics and geographic healthcare needs by analyzing current demand, supply, and service gaps. Three editions of the reports have been released—in 2012, 2018, and 2022—each revealing new projected gaps for Dubai. In summary, publishing in *Healthcare* enables these methodological lessons and policy implications to inform an international audience seeking adaptable strategies for resilient, efficient, and equitable health systems.

## 2. Research Aims and Objectives

The primary aim of this research is to analyze and compare the 2018 and 2022 Dubai Clinical Services Capacity Plan (DCSCP) reports to gain insight into the evolution of healthcare service gaps in Dubai. Additionally, this study seeks to assess the progress made in addressing these gaps, considering the developments and changes in the Emirate’s healthcare infrastructure over this period.

Objectives: The objective of this study is to evaluate the healthcare service gaps identified in the 2018 and 2022 Dubai Clinical Services Capacity Plan (DCSCP) reports. It also seeks to analyze which gaps from the 2018 report had been addressed by 2022. Furthermore, the study will identify and outline new projected gaps based on Dubai’s rapidly evolving healthcare infrastructure.

### 2.1. Research Context and Framework

Context: The Dubai Clinical Services Capacity Plan (DCSCP) 2018 and 2022 reports are in alignment with the Dubai Health Sector Strategy 2022–2026 strategic initiatives aiming to help address the healthcare service demand gaps in the Emirate of Dubai. This study used the same methodology, which includes identifying and validating the current inventory of healthcare services across nine geographical sectors of Dubai, projecting future healthcare demands up to 2030, and analyzing the supply gap. The methodology also includes identifying both short- and long-term healthcare service priorities for the Emirate [6].

### 2.2. Healthcare Service Supply and Demand Analytical Framework

This study adheres to the same eight healthcare services’ key planning units (KPUs) as illustrated in Figure 1 below. By comparing the results of the 2018 and 2022 reports based on these KPUs, we can gain a clearer understanding of which gaps have been addressed and determine whether an equilibrium between healthcare demand and supply has been achieved.

## 3. Methodology

The DCSCP 2018 and 2022 studies followed the same overall methodology to ensure consistency, including identical definitions and classifications for all eight key planning units (KPUs). The KPU’s were defined in alignment with the Dubai Health Authority (DHA) ’s planning framework, which adapts WHO Health System Building Blocks and OECD service capacity benchmarks to the Emirate’s context. These KPUs provide standardized units for comparing supply and demand across geographies and specialties, while facilitating benchmarking against international norms (OECD and WHO) [7].

For the 2018 study, all licensed healthcare facilities within Dubai’s jurisdiction—including hospitals, clinics, specialty centers, day surgery units, rehabilitation facilities, and diagnostic centers—were included. This created a full census of the healthcare sector, capturing both government and private facilities, regardless of size or scope.

For the 2022 study, only newly licensed facilities and existing facilities that had expanded their services since 2018 were surveyed. This represents a partial census, not a full enumeration of the sector. The intention was to measure incremental capacity growth between 2018 and 2022. Accordingly, the 2022 dataset does not reflect the total state of the health sector, but rather the net additions in supply during this period. Approximately 500 facilities met these criteria. Demand was projected based on age, gender, and disease-related groups (DRGs), tailored separately for UAE nationals and expatriates. DRGs were adapted from international standards, with adjustments for local differences in disease burden and socioeconomic factors. Demand estimates followed four key steps, namely Need, Demand, Activity, and Outcome, and were divided according to Dubai’s nine geographical regions [6].

Demand projections were adjusted for age, sex, and nationality using modified diagnosis-related group (DRG) classifications. This approach accounts for disease burden variability across demographic groups. The resulting service requirements were converted into per-capita metrics (e.g., beds per 10,000 residents) and benchmarked against international norms from WHO and OECD. Gaps were calculated by subtracting projected demand (adjusted by demographic weighting) from the available supply for each key planning unit (KPU), enabling comparability across years and with other health systems.

To ensure comparability across years, the same eight key planning units (KPUs)—including acute beds, outpatient rooms, ICU, and long-term care—were applied consistently, using identical definitions, data classification rules, and benchmarks. This alignment facilitated valid cross-year comparisons despite differences in data sources. However, while the 2018 study surveyed all licensed facilities across Dubai, the 2022 dataset focused specifically on newly licensed or expanded facilities (approximately 500 in total). As such, the 2022 dataset represents incremental capacity changes rather than a full sectoral census, which is important to acknowledge when interpreting year-to-year shifts.

Data and statistics: The 2018 DCSCP study collected data from all government and private healthcare facilities in Dubai. Over 3000 health facilities were surveyed within the jurisdictions of the Dubai Health Authority, Ministry of Health, Dubai Academic Healthcare Corporation, and the private sector. In contrast, the 2022 DCSCP study focused on gathering data from only newly licensed facilities, and the existing facilities who had expanded their services after 2018, totaling approximately 500 facilities. This approach provided a comprehensive view of healthcare supply in 2018 and highlighted any new additions by 2022. Both studies employed primary data collection, with all hospitals completing a survey questionnaire developed by the Dubai Health Authority [6].

## 4. Results

Table 1 illustrates changes in population size, including only includes residents living in Dubai, healthcare supply via licensed beds, and a proportional reallocation between same-day and acute overnight beds resulting from shifts in healthcare utilization patterns. Between 2018 and 2022, the population grew at an annual compound rate of 3%, while the supply of hospital beds increased by 4% annually. Similarly, the licensed bed-to-population ratio increased from 16.2 to 17.0 beds per 10,000 residents. This increase is significant, as 95% confidence intervals were calculated for the bed-to-population ratio. In 2018, there were 16.2 beds per 10,000 residents (95% CI: 15.75–16.63), while in 2022, this rose to 17.0 beds per 10,000 (95% CI: 16.56–17.42). The non-overlapping confidence intervals suggest that this change represents a statistically meaningful increase in capacity rather than random variation. Moreover, the proportion of same-day beds relative to total licensed beds rose by 4%, leading to a corresponding decrease in the share of acute overnight beds.

Table 2 shows a comparison of identified gaps across key planning units between 2018 and 2022. The analysis reveals that shortages in hospital beds (both overnight and same-day), emergency department capacity, and outpatient rooms observed in 2018 had been resolved by 2022, resulting in a surplus in these areas. Despite an addition of 76 long-term beds over the four-year period, a deficit of 138 beds persists, while the supply of ICU beds remains adequate relative to demand.

Table 3 compares the deficits in acute overnight beds across key specialties between 2018 [6] and 2022. In 2018, respiratory medicine and oncology/hematology faced a supply gap of more than 50% relative to demand [6]. By 2022, an additional 51 beds had been allocated to these specialties, resulting in a reduction in the bed deficit. The Psychiatry and Immunology and Infections specialties saw a moderate deficit (between 25% and 50%), which had been partially addressed for Psychiatry but significantly oversupplied for Immunology with the addition of 177 beds by 2022. Six specialties highlighted in green had deficits of less than 25% of total supply, some of which were partially resolved over the four years (Renal Medicine, Gastroenterology, Orthopedic), though undersupply remains significant for others, notably Cardiothoracic Surgery.

In contrast, Table 4 compares the shortages of acute same-day beds across key specialties from 2018 to 2022. In 2018, the seven specialties highlighted in red faced deficits of more than 50% relative to demand [6]. By 2022, these deficits bad been partially addressed for six specialties, with Immunology and Infection achieving a surplus with an additional 15 beds. Specialties highlighted in orange had moderate deficits in acute same-day supply, which bad been resolved by 2022. Ophthalmology, General Medicine, and Pediatric Medicine (highlighted in green) initially had deficits of less than 25% of total supply, which were resolved over the four-year period, except for Pediatric Medicine. However, it is important to note that AHRQ data on U.S. hospital stays from 1997 to 2005 highlights a significant increase in musculoskeletal surgeries, such as joint replacements and spinal procedures, contributing to higher inpatient bed utilization [8]. These findings underscore the need for strategic planning in Dubai’s healthcare infrastructure to accommodate the growing burden of orthopedic and musculoskeletal conditions, particularly with an aging population and rising lifestyle-related disorders [3].

Table 5 compares the gaps of outpatient consultation rooms across key specialties from 2018 to 2022. In 2018, the seven specialties highlighted in red faced deficits of more than 50% relative to demand [6]. By 2022, these deficits bad been partially alleviated for six specialties, while allied health achieved a surplus of 382 rooms due to increased supply. In primary care, there was a deficit of 198 rooms in 2018 [6], which bad been fully addressed within four years with the addition of 459 rooms. The specialties with minor deficits (highlighted in green) also attained sufficient supply by 2022.

## 5. Discussion

According to the Dubai Clinical Services Capacity Plan (DCSCP), in 2018 there were 5169 licensed beds available for both same-day and overnight acute care, serving a resident population of 3.2 million, which equates to 16 beds per 10,000 residents [6]. By 2022, the number of licensed beds had increased to 6030, accommodating a total population of 3.5 million [9], resulting in 17 beds per 10,000 residents. International benchmarks show that Saudi Arabia, Singapore, the UK, and the US have bed-to-population ratios ranging from 21.5 to 27.4 per 10,000 people [10]. Studies in Germany and Singapore have demonstrated that maintaining ratios above 20 beds per 10,000 is associated with reduced emergency department overcrowding and improved surgical throughput [11]. While Dubai has increased its licensed acute care beds from 16 to 17 per 10,000 residents between 2018 [6] and 2022, today it still falls short of these standards. Further investment in healthcare infrastructure may therefore be needed, but any expansion must account for Dubai’s unique demographic, including its young, transient expatriate population.

Dubai’s population and bed capacity both increased, at compound annual growth rates of 3% [9] and 4% [6], respectively. However, the ratio of acute overnight beds to same-day beds underwent significant changes over the four-year period. The proportion of same-day beds out of the total licensed beds increased by 4%, leading to a corresponding decrease in the proportion of acute overnight beds. This shift can be attributed to the transition in healthcare payment methods, where inpatient services moved to DRG-based reimbursement [12] while day-case services continued under fee-for-service arrangements. The decrease in the utilization of acute overnight beds in Dubai following DRG implementation mirrors global findings, where DRG-based payment systems have been linked to shorter lengths of stay, increased day-case surgery rates, and improved cost-efficiency without adverse patient outcomes [13]. In Germany and Australia, DRGs have also encouraged hospitals to optimize patient flow and discharge planning, reducing bottlenecks in high-demand specialties [14]. Those healthcare systems have reported that the new payment system resulted in decreased hospital activity, shorter hospital stays, and reduced inpatient costs, and led to a more data-driven decision-making process [15].

In 2018, Dubai faced a deficit in planning units like hospital beds (both overnight and same-day), emergency department capacity, and outpatient rooms [6]. Over the subsequent four years, Dubai successfully addressed these gaps through the addition of new private hospitals and clinics into the healthcare market [5]. However, the shortage of long-term care beds persists, this deficit possibly being attributed to the rising demand for long-term beds, driven by a high incidence of trauma and road accidents in Dubai [16]. Additionally, insufficient investment in expanding the supply of long-term care beds is influenced by limited financing and reimbursement mechanisms for long-term services and rehabilitation [1].

Globally, the expansion of long-term care (LTC) capacity has been supported not only by varied funding mechanisms but also by comprehensive policy structures designed to ensure sustainability and equitable access [17]. When comparing against international benchmarks, countries such as Germany, Japan, and the Netherlands have implemented systems where public or social insurance covers long-term care, funded through taxes or mandatory insurance premiums [18]. In Germany, for instance, a mandatory long-term care insurance system financed through payroll contributions is complemented by federal oversight to standardize service quality and guarantee fair access [19], whereas the United States offers a variety of public sources including Medicaid and Medicare as well as private sources through insurance and savings plans [20]. In Canada, long-term care homes are funded and regulated by the provincial government and are generally intended for individuals who can no longer live independently due to insufficient support in the community or for patients ready to be discharged from the hospital who may no longer be able to manage at home [21]. In addition, Singapore’s long-term care (LTC) homes are funded through a combination of public funding, individual payments, and insurance schemes. The government provides substantial subsidies to help make LTC services more affordable, especially for lower-income residents, with subsidies being tiered based on residents’ household income [22]. Finally, in Saudi Arabia, long-term care (LTC) homes are primarily funded by the government. The Ministry of Health (MOH) plays a central role in financing and providing healthcare services, including long-term care for the elderly and those with chronic illnesses or disabilities [23]. In summary, while Dubai has made significant progress in addressing deficits in hospital beds, emergency department capacity, and outpatient rooms over the past four years, the ongoing shortage of long-term care (LTC) beds remains a critical challenge [1]. This issue is exacerbated by the increasing demand for LTC beds, driven by high rates of trauma and road accidents [16], and by limited investment in expanding LTC infrastructure due to constrained financing and reimbursement mechanisms. These international approaches highlight the importance of stable financing models, transparent eligibility standards, and a balance between institutional and community-based care—principles that can inform Dubai’s strategy for expanding its LTC services.

Moreover, when examining the shortage of acute overnight beds by specialty, it becomes evident that there is a growing demand for respiratory, oncology, and hematology beds [6]. While there has been some increase in supply to meet these demands, there remain gaps that regulatory bodies could address by issuing licenses based on a certificate of need. This approach would target the required specialties, focusing on both infrastructure and manpower needs. According to the 2022 DCSCP report, shortages in psychiatry, pediatrics, and gastroenterology beds persist and warrant immediate attention.

Psychiatry bed shortages are not uncommon in many counties both locally and nationally. It has been stated that there is an international crisis for acute mental health services and bed capacity [24]. Based on internal benchmarks for psychiatry beds reported by the OECD [25], the U.K. and US reported 3.4 and 3.5 beds per 10,000 population, respectively, Netherlands reported 11.2 beds per 10,000, and Japan reported the highest psychiatry bed ratio at 25.8 beds per 10,000 in 2021. In contrast, Saudi Arabia and UAE have a psychiatry bed density of 1.4 and 1.8. beds per 10,000 population, respectively [26]. Emergency department visits particularly relating to patients with mental health needs, have reportedly increased in the US, Canada, UK, and Australia, resulting in longer patient waiting times [27]. In Europe, the number of inpatient psychiatric beds has decreased, intensifying pressure on available beds and altering the ward environment [28]. This shift has led to more complex challenge among admitted patients who require different levels of care compared to those patients who require specialized mental health care and are use ED beds for such purposes [27]. While most countries have reported a decrease in the number of psychiatric care beds due to the changes in the provision of care and its shift towards community-based settings, Japan “still has a very high ratio of psychiatric care beds per capita among member countries of the Organization for Economic Co-operation and Development (OECD) (p. 1)” [29]. For over 20 years, the Japanese government has implemented numerous mental health policies and laws aimed at establishing a more community-oriented mental health care system. However, progress has been slow, with more than 300,000 psychiatric patients still receiving care in hospital [29].

### 5.1. Pediatric Bed Capacity

There is a growing global demand for pediatric inpatient beds, driven by a rise in chronic diseases and increasing socioeconomic disparities [30]. Developed nations are facing an urgent need for additional pediatric beds due to factors like family structure changes and a rise in chronic illnesses among children [31], and this shortage can lead to hospital overcrowding, delayed treatments, and poorer outcomes for children requiring urgent care. Dubai, like other developed regions, is experiencing a critical gap in pediatric bed availability, with its rapidly growing population and increasing pediatric health issues highlighting the need for expanded facilities. Addressing this gap is essential to manage acute cases and provide specialized care for children with complex medical needs, a trend mirrored in healthcare systems worldwide.

### 5.2. Gastroenterology Beds

The increasing need for gastroenterology inpatient beds is a global trend, driven by the rising prevalence of GI diseases and the aging population. Globally, conditions such as inflammatory bowel disease (IBD), liver disease, gastrointestinal cancers, and other chronic digestive disorders have seen a significant rise [32]. The World Gastroenterology Organization (WGO) notes that the global burden of digestive diseases continues to grow, with conditions like IBD showing marked increases in incidence in both developed and developing regions. Moreover, lifestyle factors such as poor diet, alcohol consumption, and obesity contribute to the rising rates of gastroenterological conditions, necessitating more inpatient care facilities to manage acute episodes and complications [33]. However, despite the growing demand, there is a notable gap in the availability of specialized gastroenterology inpatient beds, particularly in developing nations and even in some regions within developed countries. This gap underscores the need for enhanced healthcare infrastructure and targeted investments to meet the global demand for specialized GI care [34,35]. Gastroenterology inpatient beds are crucial in providing specialized care for patients with complex digestive disorders, a need that has been growing in both developed and rapidly developing regions like Dubai. Despite the increasing demand, a significant gap exists due to factors such as rapid population growth, the rising prevalence of gastrointestinal conditions, and a lag in healthcare infrastructure development. Additionally, changes in diet and lifestyle, along with an aging population, have contributed to the rise in gastrointestinal diseases, further stressing the need for more specialized inpatient beds [33]. This gap highlights the need for strategic healthcare planning to expand gastroenterology services and ensure that patients receive timely and appropriate care, thus preventing complications and reducing the burden on emergency services.

## 6. Conclusions

Since the previous DSCSP report in 2018, there has been a significant growth in hospitals and outpatient facilities leading to an increase in capacity for the defined KPUs including medical equipment and healthcare workforce. This gain in licensed bed capacity is statistically significant, as confirmed by confidence intervals for the bed-to-population ratio. The observed improvement is not only policy-relevant but also statistically robust, strengthening confidence in the healthcare system’s progress over the four-year period. However, because of the continuing population growth there remain significant shortages (gaps) in certain service lines.

Comprehensive policy solutions to address these gaps include targeted investments in long-term care and specialty areas like psychiatry, cardiothoracic surgery, and pediatric medicine, where deficits persist. Optimizing the use of same-day beds and improving referral networks can address and enhance bed utilization efficiency. Implementing flexible infrastructure planning with real-time monitoring of bed capacity would facilitate timely adjustments based on demand, especially during emergencies. Additionally, investing in preventive care and early interventions could reduce the strain on acute care beds, while standardizing bed definitions and improving resource allocation would ensure a more balanced and responsive healthcare system.

### 6.1. Policy Implications

Effective healthcare capacity planning is vital to maintaining a resilient, efficient, and responsive healthcare system that can adapt to both routine and extraordinary demands. Failing to adopt such policy options has the potential of contributing to oversupply, creating an inefficient and costly healthcare system unable to meet growing and evolving patient needs.

### 6.2. Study Strengths and Limitations

The strength of this study lies in its use of novel, carefully collected data, combined with robust statistical analysis and thorough interpretation of the results. Additionally, the study employed a full enumeration approach (census) to capture data on healthcare supply, gathering information from every licensed and operational facility in Dubai.

Limitations: This study compares datasets from only two reports and relies on static supply figures, limiting the precision of forecasting. While consistent KPUs and classifications were used, the 2018 dataset included all licensed facilities (a full census), whereas the 2022 dataset covered only newly licensed or expanded facilities. As such, findings reflect incremental growth rather than total sector capacity growth and should be interpreted accordingly. Additionally, key performance indicators—such as bed occupancy, average length of stay (ALOS), and patient throughput—were not included. Future research should incorporate these metrics alongside structural supply–demand analysis to offer a more comprehensive assessment of system performance.

## Figures and Tables

**Figure 1 healthcare-13-02127-f001:**
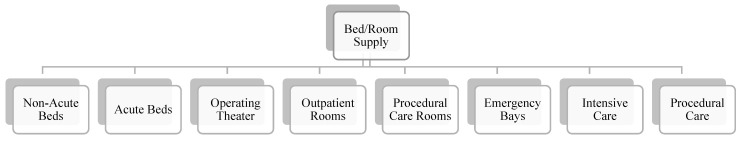
Health care service key planning units (KPUs) [6].

**Table 1 healthcare-13-02127-t001:** Dubai population [2] and licensed beds in Dubai (2018 and 2022).

	2018	2022	* CAGR
Population	3,192,275	3,549,900	3%
Licensed beds	5169	6030	4%
Proportion of same days beds	16%	20%	-
Proportion of acute overnight beds	84%	80%	-

* Compound Annual Growth Rate.

**Table 2 healthcare-13-02127-t002:** Key planning unit gaps in 2018 and 2022.

Key Planning Units (KPU)	2018	2022
Acute Overnight Bed Gap	−239	1728
Acute Same Day Gap	−364	294
Outpatient Room Gap	−1769	4707
Emergency Department	−100	182
Adult ICU	484	691
PICU	49	73
NICU	68	211
Non-Acute Beds/Long Term Beds	−214	−138
OT	70	129

**Table 3 healthcare-13-02127-t003:** Acute overnight bed gaps by specialty for 2018 [6] and 2022.

Acute Overnight Bed Gap-Specialty	2018–2020	2022
Respiratory Medicine	−108	−67
Oncology and Hematology	−43	−25
Psychiatry	−78	−48
Immunology and Infections	−61	116
Renal Medicine	−10	5
Gastroenterology	−10	20
Pediatric Surgery	−3	−4
Pediatric Medicine	−5	−17
Orthopedics and Rheumatology	−1	73
Cardiothoracic Surgery	0	−25

Red: High > 50% deficit in total supply gap. Yellow: Medium 25–50% deficit in total supply gap. Green: Low 0–25% deficit in total supply gap.

**Table 4 healthcare-13-02127-t004:** Acute same-day bed gaps by specialty for 2018 [6] and 2022.

Acute Same-Day Bed Gap-Specialty	2018–2020	2022
Hematology and Oncology	−44	−33
Dentistry	−15	−11
Renal Medicine	−8	−3
Gastroenterology	−22	−8
Pediatric Surgery	−18	−16
Immunology and Infections	−2	13
Psychiatry	−6	−5
Dialysis	−39	3
General Surgery	−25	19
Ophthalmology	−4	2
General Medicine	−24	72
Pediatric Medicine	−15	−18

Red: High > 50% deficit in total supply gap. Yellow: Medium 25–50% deficit in total supply gap. Green: Low 0–25% deficit in total supply gap.

**Table 5 healthcare-13-02127-t005:** Outpatient consultation room gaps by specialty for 2018 [6] and 2022.

Outpatient Consultation Rooms-Specialty	2018–2020	2022
Respiratory Medicine	−100	−67
Trauma and Injury	−20	−15
Neonatology	−4	4
Allied Health	−120	382
Neurology	−75	−31
Immunology and Infections	−20	−11
Oncology and Hematology	−35	−20
Endocrinology	−46	31
Primary Care	−198	261
Vascular Surgery	−2	29
Renal Medicine	−3	11
Gastroenterology	−8	35

Red: High > 50% deficit in total supply gap. Yellow: Medium 25–50% deficit in total supply gap. Green: Low 0–25% deficit in total supply gap.

## Data Availability

All data generated or analyzed during this study are included in this published article, the Dubai Clinical Services Capacity Plan 2018–2030, and can be accessed via the following: https://www.dha.gov.ae/uploads/122021/e9b6b25d-1339-4f2e-8fbf-2b8ea3217315.pdf (accessed on 20 April 2024).

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
