# Peer review of "Healthcare Service Gap Analysis: A Comparison of Trend Data from 2018 and 2022 Dubai Clinical Services Capacity Reports"

_healthcare, 2025, doi:10.3390/healthcare13172127_

Round 1

Reviewer 1 Report

Comments and Suggestions for Authors

Dear Authors,

We would like to congratulate you on your comprehensive and well-organized manuscript analyzing the evolution of healthcare service capacity in Dubai between 2018 and 2022. Your work provides a valuable, data-driven insight into how healthcare systems can adapt to rapid population and infrastructure changes.

To further strengthen the manuscript, we respectfully offer the following suggestions:

- Contextual Framing: While your introduction thoroughly describes the local healthcare context, it would benefit from a brief overview of similar strategic capacity planning approaches in other international settings. This would provide comparative grounding for your findings and reinforce the relevance of the Dubai experience on a global scale.

- Methodological Clarification: Please consider including additional detail about how consistency in the application of KPUs across 2018 and 2022 was ensured. Also, while the 2018 dataset included all licensed facilities, the 2022 sample focused only on new or expanded ones. This methodological distinction could affect comparability and should be explicitly addressed in the limitations.

- Discussion Section: Your analysis of persistent service gaps is strong. We suggest deepening the discussion on psychiatric and pediatric care needs with recent global trends and referencing similar capacity constraints internationally. For long-term care, we encourage further elaboration on the financing mechanisms and policy frameworks that enable successful expansion in benchmark countries (e.g., Germany, Japan, Canada), beyond funding descriptions.

- References: The references are generally adequate and include recent Dubai-specific sources. We recommend incorporating more peer-reviewed international academic literature in sections that analyze comparative healthcare performance and planning metrics.

We appreciate the rigor and relevance of your work and are confident that these minor revisions will enhance its impact and scholarly contribution.

Author Response

Healthcare Services Gap Analysis: A Comparison between 2018 and 2022 Dubai Clinical Services Capacity Report

Response to: Author's Reply to the Review Report (Reviewer 1)

1. Summary

Thank you very much for taking the time to review this manuscript. Please find detailed responses below and the corresponding revisions/corrections highlighted/in track changes in the re-submitted files.

2. Point-by-point response to Comments and Suggestions for Authors

Comments 1: Contextual Framing: While your introduction thoroughly describes the local healthcare context, it would benefit from a brief overview of similar strategic capacity planning approaches in other international settings. This would provide comparative grounding for your findings and reinforce the relevance of the Dubai experience on a global scale.

Response 1: Thank you for pointing this out. We agree with this comment. Therefore, we have included an additional paragraph starting on page 3 in the manuscript. Please see the comments section for the amended section.

Comments 2: Methodological Clarification: Please consider including additional detail about how consistency in the application of KPUs across 2018 and 2022 was ensured. Also, while the 2018 dataset included all licensed facilities, the 2022 sample focused only on new or expanded ones. This methodological distinction could affect comparability and should be explicitly addressed in the limitations.

Response 2: Agree. We have, accordingly, included additional detail on the methodological distinction. You can find this additional paragraph starting on page 6, see comment section that highlights this paragraph.

Comments 3: Discussion Section: Your analysis of persistent service gaps is strong. We suggest deepening the discussion on psychiatric and pediatric care needs with recent global trends and referencing similar capacity constraints internationally. For long-term care, we encourage further elaboration on the financing mechanisms and policy frameworks that enable successful expansion in benchmark countries (e.g., Germany, Japan, Canada), beyond funding descriptions.

Response 3: We have included Germany and Canada was already referenced before. We provided in. We have, accordingly, included additional detail on the methodological distinction. You can find this additional paragraph starting on page 6, see comment section that highlights this paragraph.

Reviewer 2 Report

Comments and Suggestions for Authors

  1. Abstract:

- References are cited in the Abstract. All must be removed

- The results mention "statistically significant improvement" without specifying which outcomes improved, how much, or the statistical tests used.

  1. Introduction: Cited references begin from 2 (and not 1). They are not in order

  1. Aims and Objectives:

Under Objectives sub-heading (line 84) = “The aim of this study is to evaluate the healthcare service gaps identified”  - Aim should be replaced by “objectives”

Line 88-89 = These objectives will offer a comprehensive understanding of how healthcare needs and services have changed over time in the Emirate. = It should be deleted, as these are study implications

  1. Methods:

-Mention in detail the inclusion criteria for 2018 and 2022 survey and how both differed from each other

  1. Results:

- In Table 1 – Whether the population mentioned also includes tourists/people from other nations who came to Dubai for education/occupation or just the citizens?

- It will be good to indicate below the tables, the importance of color coding (Table 3, 4, 5)

  1. References:

- Most references don’t have Page numbers. They must be written as per the journal style

Author Response

Response to: Author's Reply to the Review Report (Reviewer 2)

1. Summary

Thank you very much for taking the time to review this manuscript. Please find detailed responses below and the corresponding revisions/corrections highlighted/in track changes in the re-submitted files.

2. Point-by-point response to Comments and Suggestions for Authors

Comments 1: References are cited in the Abstract. All must be removed

Response 1: Thank you for pointing this out. All references are removed.

Comments 2: The results mention "statistically significant improvement" without specifying which outcomes improved, how much, or the statistical tests used.

Response 2: Sure. We have updated the section. Please see the results section in the abstract which has been amended.  Please see page #2.

Comments 3:  Introduction: Cited references begin from 2 (and not 1). They are not in order

Response 3: Fixed.

Comments 4: Aims and Objectives: Under Objectives sub-heading (line 84) = “The aim of this study is to evaluate the healthcare service gaps identified”  - Aim should be replaced by “objectives”. Line 88-89 = These objectives will offer a comprehensive understanding of how healthcare needs and services have changed over time in the Emirate. = It should be deleted, as these are study implications

Response 4: Thank you for pointing this out. The term “aim” is replaced with objective and the last sentence of the paragraph is deleted as per your suggestions.

Comments 5: Methods: Mention in detail the inclusion criteria for 2018 and 2022 survey and how both differed from each other

Response 5: Noted. We have updated the methodology section to specify the survey difference for 2018 and 2022 survey.

Comments 6: In Table 1 – Whether the population mentioned also includes tourists/people from other nations who came to Dubai for education/occupation or just the citizens?

- It will be good to indicate below the tables, the importance of color coding (Table 3, 4, 5)

Response 6: The population includes all residents of the emirate and not tourists. Please see page 7. Also, with regards to the table and importance of the color coding labels are mentioned already, please see page 9.

Comments 7: Most references don’t have Page numbers. They must be written as per the journal style

Response 7: Amended

Reviewer 3 Report

Comments and Suggestions for Authors

I attach my comments. 

Comments on the Quality of English Language

English language requires editing. Several sentences are lengthy or ambiguous. A professional edit is recommended.

References are inconsistently formatted (e.g., [2], [3]) and must follow MDPI’s citation style precisely.

Figures and Tables are too dense. Suggest:

    • Moving Tables 3–5 to Supplementary Material;
    • Avoiding duplication between text and tables.

Author Response

Healthcare Services Gap Analysis: A Comparison between 2018 and 2022 Dubai Clinical Services Capacity Report

Response to: Author's Reply to the Review Report (Reviewer 3)

1. Summary

Thank you very much for taking the time to review this manuscript. Please find detailed responses below and the corresponding revisions/corrections highlighted/in track changes in the re-submitted files.

Scientific Rationale and Contribution

Q1. The introduction provides descriptive context but does not identify a clear research gap, scientific question, or theoretical basis. The study lacks a hypothesis-driven framework.

A.    Response on Page. 3

Q2. The manuscript should be reframed to demonstrate what the Dubai case contributes to global knowledge about health system planning. How can the findings inform or be compared to international experiences in rapidly developing or demographically unique settings?

A.     Response on Page 4

Q3. The authors should clearly justify why this study is suitable for an international audience and explain how it contributes to the comparative health systems literature (e.g., by utilizing models from the WHO or OECD).

A.    Response on Page 5

Q4. The comparative analysis across two timepoints is descriptive. To meet publication standards in Healthcare, it must evolve into an analytically grounded, policy-informing paper with transferable implications.

A.    We agree that papers with strong policy insights are important for the journal. However, this paper was intentionally designed as a descriptive comparison between two official government studies (DCSCP 2018 and 2022), which were conducted to support healthcare planning in Dubai.Our aim was to document how healthcare supply and demand have changed over time and to highlight areas where progress has been made and where gaps still remain. While we do not use inferential statistics, the analysis is grounded in a real-world planning framework that includes standardized metrics, population-adjusted demand modeling, and alignment with WHO and OECD benchmarks. This offers practical insights for policymakers without going into more advanced statistical modeling.

We believe the descriptive approach still provides value—especially for health systems that are growing rapidly, rely heavily on private sector providers, and are facing similar demographic pressures. For this reason, we would prefer to keep the current structure, as it reflects the original intent of the study and stays true to the data used.

Use of Subsections and Formatting

Q1. The inclusion of subheadings like 4.1, 4.2, 5.1, 5.2 is inconsistent with MDPI formatting for original research articles unless methodologically justified. These sections resemble internal planning documents rather than peer-reviewed

research.

A.    Not sure where these subheadings are reflected? Please clarify

Q2. Recommend restructuring the Discussion section thematically and integrating

content without numerical subsections.

A.    No numerical subsections are in the discussions sections, they are just broken down into gaps we are focused on.

Methodological Clarity: data collection process, methodology, sensitivity analysis.

Q1. The 2018 dataset was a complete census (>3,000 facilities), whereas 2022 included only new or expanded facilities (~500), leading to potential sampling asymmetry and bias.

A.    Please see page 6, for updated methodology clarification.

Q2. Were the same data definitions, categories, and demand estimation approaches applied consistently across both years?

A.     Yes, The DCSCP 2018 and 2022 studies followed the same overall methodology to ensure consistency, including identical definitions and classifications for all eight Key Planning Units (KPUs). See page 6.

Q3. The methodology lacks clarity on how were established. Are these based on WHO, OECD, or DHA-defined criteria? The rationale must be specified.

A.    Please see page 6, for updated methodology clarification.

Q4. No inferential analysis or sensitivity testing results are presented, despite being mentioned. If sensitivity analysis was performed, results must be included. Otherwise, the claim should be removed.

A.    The methodology section has been amended including the data collection. The sensitivity analysis section has been removed.

Limitations 

Q5. The manuscript present a limitations section, but it is not essential.

A.    Limitations section has been amended

Ethical and Governance Considerations

Q6. Although the study does not involve human participants, it uses institutional data. The authors should explicitly state: Whether ethical review or governance board approval was sought or required. If data use was approved by Dubai Health Authority and under what

conditions.

A.     As per the DHA guidelines, since there are no human subjects, ethical review or governance board approval was not required.

Gap Analysis Validity and Comparability

Q6. The term gap is used repeatedly without standardized metrics (e.g., per 10,000 residents, adjusted by age/sex/disease burden). Absolute numbers are less meaningful without context

A.    In estimating demand, we took into account differences in age, gender, and nationality by using a method based on diagnosis-related groups (DRGs). This helped us better reflect the actual healthcare needs of Dubai’s diverse population. We then calculated how many services—such as hospital beds or outpatient rooms—were needed per 10,000 people. These numbers were compared with what is currently available to identify whether there was a shortage or surplus. This approach allowed us to make fair comparisons between 2018 and 2022, and also to compare Dubai’s situation with international standards, such as those from the WHO and OECD. See amended section page 7

Q7. The results should include international comparators (e.g., beds per capita vs.

WHO thresholds, OECD data) to improve generalizability and interpretation.

A.    See amended section page 7

Q8. The manuscript does not account for key healthcare performance indicators such as bed occupancy rates, average length of stay, or patient throughput all critical for understanding capacity adequacy.

A.    Although both the 2018 and 2022 Dubai Clinical Services Capacity Plan (DCSCP) reports include operational performance indicators—such as average bed occupancy rates, average length of stay (ALOS), and patient throughput—this study focuses exclusively on comparing structural capacity gaps across Key Planning Units (KPUs).

Round 2

Reviewer 2 Report

Comments and Suggestions for Authors

The revised version of the article looks significantly improved. We can proceed ahead with acceptance

Author Response

The revised version of the article looks significantly improved. We can proceed ahead with acceptance. 

A. Many thank for you advice and guidance. 

Reviewer 3 Report

Comments and Suggestions for Authors

Thank you for submitting the revised manuscript. I appreciate the considerable effort undertaken to address my comments and reposition the manuscript toward international relevance.

Positive Developments:

  • The manuscript now includes an expanded Introduction that situates Dubai's healthcare strategy within a global context. The use of international case examples (e.g., UK, Germany, Singapore) enhances the relevance and utility of the findings for a broader audience.

  • Methodological clarity has improved, especially in regard to the definition and application of Key Planning Units (KPUs), the distinction between the 2018 and 2022 datasets, and the use of demographic-adjusted demand projections.

  • The structure and presentation have been refined in accordance with MDPI guidelines, particularly through the removal of numbered internal subsections and improved formatting of the Discussion section.

Areas for Further Consideration:

  • While the explanation of the 2022 partial census is adequate, the manuscript would benefit from explicit clarification that the results should be interpreted as reflecting incremental capacity growth, not full sectoral trends.

  • The manuscript still lacks quantitative inferential analysis, which could strengthen the credibility of its conclusions. Even simple statistical comparisons (e.g., effect sizes or confidence intervals) would help validate the observed trends.

  • Key healthcare system performance indicators (e.g., bed occupancy rates, average length of stay, throughput) are not addressed. Their inclusion, even briefly in the Limitations section, would improve the analytic rigor.The ethical governance section is vague. It would be advisable to explicitly state whether data use approval was formally granted by the Dubai Health Authority or another relevant body.

Overall, the manuscript has significantly improved in scientific framing and international relevance. It still reads somewhat descriptively, but it now presents valuable comparative planning insights that merit publication following minor additional clarifications.

Author Response

Areas for Further Consideration:

  • While the explanation of the 2022 partial census is adequate, the manuscript would benefit from explicit clarification that the results should be interpreted as reflecting incremental capacity growth, not full sectoral trends.

A. Noted. We have amended this section. Please see page 4, starting at line 141.

  • The manuscript still lacks quantitative inferential analysis, which could strengthen the credibility of its conclusions. Even simple statistical comparisons (e.g., effect sizes or confidence intervals) would help validate the observed trends.

A. Noted. We have included some inferential analysis on line 180 & 370.

  • Key healthcare system performance indicators (e.g., bed occupancy rates, average length of stay, throughput) are not addressed. Their inclusion, even briefly in the Limitations section, would improve the analytic rigor. The ethical governance section is vague. It would be advisable to explicitly state whether data use approval was formally granted by the Dubai Health Authority or another relevant body.

A. Noted. Please see updated ethical governance section, line 112. In addition , please also see updated limitations section, line 396.